# Design, Synthesis, and Anti-Melanogenic Activity of 2-Mercaptomethylbenzo[*d*]imidazole Derivatives Serving as Tyrosinase Inhibitors: An In Silico, In Vitro, and In Vivo Exploration

**DOI:** 10.3390/antiox13101248

**Published:** 2024-10-16

**Authors:** Hee Jin Jung, Hyeon Seo Park, Hye Jin Kim, Hye Soo Park, Yujin Park, Pusoon Chun, Hae Young Chung, Hyung Ryong Moon

**Affiliations:** 1Department of Manufacturing Pharmacy, College of Pharmacy and Research Institute for Drug Development, Pusan National University, Busan 46241, Republic of Korea; hjjung2046@pusan.ac.kr (H.J.J.); gustj6956@pusan.ac.kr (H.S.P.); khj3358@pusan.ac.kr (H.J.K.); hyesoo0713@pusan.ac.kr (H.S.P.); 2Department of Medicinal Chemistry, New Drug Development Center, Daegu-Gyeongbuk Medical Innovation Foundation, Daegu 41061, Republic of Korea; pyj1016@kmedihub.re.kr; 3College of Pharmacy and Inje Institute of Pharmaceutical Sciences and Research, Inje University, Gimhae 50834, Republic of Korea; pusoon@inje.ac.kr; 4Department of Pharmacy, College of Pharmacy and Research Institute for Drug Development, Pusan National University, Busan 46241, Republic of Korea; hyjung@pusan.ac.kr

**Keywords:** tyrosinase, zebrafish larvae, melanin, 2-mercaptomethylbenzo[*d*]imidazole, copper ion, chelation

## Abstract

2-Mercaptomethylbenzo[*d*]imidazole (2-MMBI) derivatives were designed and synthesized as tyrosinase (TYR) chelators using 2-mercaptomethylimidazole scaffolds. Seven of the ten 2-MMBI derivatives exhibited stronger inhibition of mushroom TYR activity than kojic acid. Their ability to chelate copper ions was demonstrated through experiments using the copper chelator pyrocatechol violet and assays measuring TYR activity in the presence or absence of exogenous CuSO_4_. The inhibition mechanisms of derivatives **1**, **3**, **8**, and **9**, which showed excellent TYR inhibitory activity, were elucidated through kinetic studies and supported by the docking simulation results. Derivatives **3**, **7**, **8**, and **10** significantly inhibited cellular TYR activity and melanin production in B16F10 cells in a dose-dependent manner, with stronger potency than kojic acid. Furthermore, in situ, derivatives **7** and **10** showed stronger inhibitory effects on B16F10 cell TYR activity than kojic acid. Six derivatives, including **8**, showed highly potent depigmentation in zebrafish larvae, outpacing kojic acid even at 200–670 times lower concentrations. Additionally, all derivatives could scavenge for reactive oxygen species without causing cytotoxicity in epidermal cells. These results suggested that 2-MMBI derivatives are promising anti-melanogenic agents.

## 1. Introduction

Melanin is a biopolymeric dark pigment that protects human skin from ultraviolet (UV) rays by absorbing them [1]. This pigment is found in all living organisms, including bacteria, fungi, plants, and humans [2,3]. In humans, melanin serves as a major factor in determining the skin phenotype [4]. Melanin also affects the colors of hair and pupils [4]. In humans, melanin is divided into three distinct types: pheomelanin (red to yellow), eumelanin (black to brown), and neuromelanin. The first two types are found in the skin, whereas the third type is found in the brain. Melanin pigment is biosynthesized in melanosomes, which are organelles within melanocytes present in the epidermis, in response to various factors including exposure to UV rays, α-melanocyte-stimulating hormones (α-MSHs), keratinocyte growth factors, stem cell factors, and radical sources [5]. Melanin synthesis, also known as melanogenesis, is a complex process involving various enzymatic and chemical reactions [6]. Abnormal melanin formation in the body can cause esthetic problems and pigment-related disorders such as lentigines, melasma, cervical poikiloderma, age spots, pityriasis alba, vitiligo, melanoma, incontinentia pigmenti, chloasma, freckles, progressive pigmentary purpura, albinism, and even skin cancer [7,8]. Additionally, the browning of vegetables, fruits, and fungi, such as mushrooms, is caused by melanin accumulation through melanin production, which has a significant impact on crop quality [9]. Furthermore, melanin has been reported to be associated with neurodegenerative diseases, including Huntington’s, Parkinson’s, and Alzheimer’s diseases [10,11,12,13,14]. Currently, only a handful of products, such as arbutin, kojic acid, and hydroquinone, are used clinically as cosmetic agents for skin whitening or as treatments for hyperpigmentation-related disorders; however, these products have various side effects, including bone marrow and renal toxicity, contact dermatitis, skin irritation, carcinogenicity, and inadequate potency [15,16,17,18]. Therefore, considerable attention should be paid to the development of newer, safer, and more effective anti-melanogenic agents.

Enzymes, such as tyrosinase (TYR), TYR-related protein-1 (TRP-1), and TRP-2 are involved in melanogenesis [19]. TYR is involved in the initial two-step process of melanogenesis through its monophenolase and diphenolase activities, and it serves as the rate-determining enzyme in melanogenesis [20,21]. l-Tyrosine is transformed into dopaquinone by the successive action of TYR, which is a common intermediate in the biosynthesis of pheomelanin and eumelanin [22]. TRP-1 and TRP-2 are required for the biosynthesis of eumelanin, not of pheomelanin [23]. Eumelanin is a water-insoluble, black-to-brown polymeric pigment, whereas pheomelanin is a water-soluble, sulfur-containing, yellow-to-red polymeric pigment. Pheomelanin sulfur is derived from l-cysteine and glutathione. TYR is a polyphenol oxidase containing two copper ions within its active site [24,25] and is the main enzyme responsible for the browning of crops, such as vegetables and fruits, and for human skin hyperpigmentation [26]. Thus, it has received steady attention as a key molecular target for treating diseases related to abnormal skin hyperpigmentation and for inhibiting browning in crops. TYR inhibitors (TYRIs) are broadly divided into three types based on their inhibition mechanisms: competitive inhibitors, noncompetitive inhibitors, and mixed inhibitors, which have the properties of both competitive and noncompetitive inhibitors. Both competitive inhibitors and mixed inhibitors compete with TYR substrates for active sites. There are two types of TYRIs that compete with TYR substrates at active sites: TYRIs that interact only with the amino acid residues present in the TYR active site and TYRIs that chelate with the copper ions present in the TYR active site. Kojic acid and phenylthiourea (PTU) belong to the latter [27,28].

Recently, we reported that 2-mercaptobenzo[*d*]imidazole and 2-mercaptobenzo[*d*]thiazole derivatives, as depicted in Figure 1, are potent TYRIs [29,30,31]. As an extension of our study on novel TYR chelators, we designed 2-mercaptomethylimidazole as a potential scaffold capable of chelating copper ions by forming a five-membered ring with them, and we synthesized its derivatives, 2-mercaptomethylbenzo[*d*]imidazoles (2-MMBIs) **1**–**10**, as potential TYRIs (Figure 1). To confirm the anti-TYR activity, the inhibitory effect of these derivatives on TYR activity was investigated using mushroom TYR and B16F10 murine cells and reconfirmed using an in situ B16F10 cellular TYR activity assay. Moreover, to confirm the melanogenesis-inhibitory activities of these derivatives, the melanin levels in B16F10 cells and in vivo depigmentation efficacy in zebrafish larvae were investigated. Furthermore, the copper-chelating abilities of these derivatives were evaluated experimentally by using pyrocatechol violet, a known copper-chelating agent, and by comparing the mushroom TYR activity in the presence or absence of exogenous copper ions. Reports that antioxidant activity is closely related to the inhibition of melanogenesis have led to the measurement of the antioxidant activities of these derivatives.

## 2. Materials and Methods

### 2.1. Synthesis

#### 2.1.1. General Methods

The reaction was monitored using thin-layer chromatography (Merck Silica gel 60 F_254_). The chemicals and solvents were acquired from Thermo Fisher Scientific (Carlsbad, CA, USA) and SEJIN CI Co. (Seoul, Republic of Korea). The reaction mixture was purified by washing the filter cake with appropriate solvents or by performing silica gel (MP Silica 60 Å) column chromatography. Nuclear magnetic resonance (NMR) (^1^H [400 MHz] and ^13^C [100 MHz]) spectra were acquired using a JEOL ECZ400S spectrometer (Tokyo, Japan). CDCl_3_ and dimethyl sulfoxide-*d*_6_ (DMSO-*d*_6_) were used as NMR solvents, and the coupling constant (*J*) and chemical shift (*δ*) were expressed in Hz and ppm, respectively. ^1^H and ^13^C NMR spectra of all compounds are provided in the Appendix A.

#### 2.1.2. Synthesis of 4-(Trifluoromethyl)benzene-1,2-diamine (**11**)

Water (4 mL) was added to a suspension solution of 2-nitro-4-(trifluoromethyl)aniline (250 mg, 1.21 mmol), iron (271 mg, 4.85 mmol), and NH_4_Cl (32 mg, 0.60 mmol) in 1,4-dioxane (4 mL). The reaction mixture was refluxed for 1.5 h and cooled to room temperature. After filtration through a pad of Celite^®^, the filtrate evaporated under reduced pressure, and the resulting residue was partitioned between water and ethyl acetate. The organic layer was first dried over anhydrous MgSO_4_, filtered, and then evaporated under reduced pressure. The resulting residue was purified by silica gel column chromatography using hexane:ethyl acetate (1.3:1) as the eluent, yielding 4-(trifluoromethyl)benzene-1,2-diamine (**11**) (176 mg, 82%).

^1^H NMR (CDCl_3_, 400 MHz): *δ* 6.99–6.96 (m, 1H, 5-H), 6.92 (d, 1H, *J* = 2.0 Hz, 3-H), 6.71 (d, 1H, *J* = 8.4 Hz, 6-H); ^13^C NMR (CDCl_3_, 100 MHz): *δ* 138.3, 134.1, 124.8 (q, *J* = 269.2 Hz), 121.9 (q, *J* = 32.1 Hz), 117.7 (q, *J* = 4.0 Hz), 115.6, 113.6 (q, *J* = 3.9 Hz).

#### 2.1.3. General Preparation of 2-MMBI Derivatives **1**–**10**

A solution of 1,2-phenylenediamine (150 mg; 1,2-phenylenediamine for **1**, 4-chlorophenylenediamine for **2**, 4-methoxyphenylenediamine for **3**, 4-fluorophenylenediamine for **4**, 4-cyanophenylenediamine for **5**, 4-nitrophenylenediamine for **6**, 4-trifluoromethylphenylenediamine for **7**, 4-methylphenylenediamine for **8**, 3-methylphenylenediamine for **9**, and 4,5-dimethylphenylenediamine for **10**) and thioglycolic acid (2.0 equiv.) in 4N-HCl solution (3.5 mL) was refluxed for 5–15 h. After cooling, the reaction mixture was treated with a 2N-NaOH solution to adjust the pH to 7.5 and was partitioned between ethyl acetate and water. The organic layer was dried over anhydrous MgSO_4_, filtered, and then evaporated under reduced pressure. The resulting residue was purified using silica gel column chromatography using dichloromethane and methanol (20:1–25:1) as eluents to obtain **2**, **3,** and **5**–**10** as solids. To purify derivatives **1** and **4**, the pH was adjusted to 7.5 with a 2N-NaOH solution, water was added, and filtration was performed to obtain the target derivatives as solids: derivatives **1** and **4** were obtained as solids in 63% and 98% yields, respectively. All derivatives had purities of 98% or greater.

(1*H*-Benzo[*d*]imidazol-2-yl)methanethiol (**1**) [32]

^1^H NMR (DMSO-*d*_6_, 400 MHz): *δ* 12.78 (brs, 1H, NH), 7.55–7.46 (m, 2H, 5-H, 6-H), 7.18–7.12 (m, 2H, 4-H, 7-H), 3.92 (s, 2H, C*H*_2_); ^13^C NMR (DMSO-*d*_6_, 100 MHz): *δ* 154.2, 139.3, 122.1, 115.4, 36.2; yield, 63%.

(5-Chloro-1*H*-benzo[*d*]imidazol-2-yl)methanethiol (**2**) [33]

^1^H NMR (CDCl_3_, 400 MHz): *δ* 9.41 (brs, 1H, NH), 7.62 (d, 1H, *J* = 2.0 Hz, 4-H), 7.53 (d, 1H, *J* = 8.4 Hz, 7-H), 7.25 (dd, 1H, *J* = 8.4, 2.0 Hz, 6-H), 4.04 (s, 2H, C*H*_2_); ^13^C NMR (CDCl_3_, 100 MHz): *δ* 152.2, 139.2, 137.0, 128.9, 123.8, 115.9, 115.1, 36.6; yield, 68%.

(5-Methoxy-1*H*-benzo[*d*]imidazol-2-yl)methanethiol (**3**)

^1^H NMR (CDCl_3_, 400 MHz): *δ* 8.42–7.60 (brs, 1H, NH), 7.51 (d, 1H, *J* = 8.8 Hz, 7-H), 7.07 (d, 1H, *J* = 2.0 Hz, 7-H), 6.90 (dd, 1H, *J* = 8.8, 2.0 Hz, 6-H), 3.99 (s, 2H, C*H*_2_), 3.81 (s, 3H, OC*H*_3_); ^13^C NMR (CDCl_3_, 100 MHz): *δ* 156.8, 150.6, 138.5, 133.7, 116.2, 112.7, 97.4, 55.9, 36.6; yield, 54%.

(5-Fluoro-1*H*-benzo[*d*]imidazol-2-yl)methanethiol (**4**) [34]

^1^H NMR (DMSO-*d*_6_, 400 MHz): *δ* 7.49 (dd, 1H, *J* = 8.8, 4.8 Hz, 7-H), 7.30 (dd, 1H, *J* = 9.6, 2.4 Hz, 4-H), 6.99 (dd, 1H, *J* = 11.2, 8.8, 2.4 Hz, 6-H), 4.14 (s, 2H, C*H*_2_); ^13^C NMR (DMSO-*d*_6_, 100 MHz): *δ* 159.0 (C5, d, *J* = 233.8 Hz), 152.6 (C2), 139.6, 135.7, 116.1, 110.5 (d, *J* = 25.4 Hz), 101.5 (d, *J* = 24.2 Hz), 36.0; yield, 98%.

2-(Mercaptomethyl)-1*H*-benzo[*d*]imidazole-5-carbonitrile (**5**)

^1^H NMR (DMSO-*d*_6_, 400 MHz): *δ* 12.89 (brs, 1H, NH), 8.07 (brd, 1H, *J* = 1.2 Hz, 4-H), 7.68 (d, 1H, *J* = 8.4 Hz, 7-H), 7.57 (dd, 1H, *J* = 8.4, 1.2 Hz, 6-H), 4.23 (s, 2H, C*H*_2_); ^13^C NMR (DMSO-*d*_6_, 100 MHz): *δ* 154.7, 141.4, 139.6, 126.3, 126.0, 120.2, 116.3, 105.0, 36.3; yield, 69%.

(5-Nitro-1*H*-benzo[*d*]imidazol-2-yl)methanethiol (**6**) [35,36]

^1^H NMR (DMSO-*d*_6_, 400 MHz): *δ* 13.09 (brs, 1H, NH), 8.40 (d, 1H, *J* = 2.0 Hz, 4-H), 8.09 (dd, 1H, *J* = 8.8, 2.0 Hz, 6-H), 7.68 (d, 1H, *J* = 8.8 Hz, 7-H), 4.26 (s, 2H, C*H*_2_); ^13^C NMR (DMSO-*d*_6_, 100 MHz): *δ* 155.9, 143.0, 142.8, 139.8, 118.3, 115.1, 112.3, 36.0; yield, 72%.

(5-(Trifluoromethyl)-1*H*-benzo[*d*]imidazol-2-yl)methanethiol (**7**)

^1^H NMR (DMSO-*d*_6_, 400 MHz): *δ* 15.62 (brs, 1H, NH), 7.89 (d, 1H, *J* = 1.6 Hz, 4-H), 7.71 (d, 1H, *J* = 8.4 Hz, 7-H), 7.49 (dd, 1H, *J* = 8.4, 1.6 Hz, 6-H), 4.24 (s, 2H, C*H*_2_); ^13^C NMR (DMSO-*d*_6_, 100 MHz): *δ* 154.2, 141.0, 139.6, 125.6 (5-*C*F_3_, q, *J* = 270.2 Hz), 123.0 (C5, q, *J* = 31.1 Hz), 119.1 (q, *J* = 2.8 Hz), 115.7, 113.6, 35.9; yield, 52%.

(5-Methyl-1*H*-benzo[*d*]imidazol-2-yl)methanethiol (**8**) [37]

^1^H NMR (CDCl_3_, 400 MHz): *δ* 8.62 (brs, 1H, NH), 7.52 (d, 1H, *J* = 8.4 Hz, 7-H), 7.42 (s, 1H, 4-H), 7.09 (d, 1H, *J* = 8.4 Hz, 6-H), 4.02 (s, 2H, C*H*_2_), 2.46 (s, 3H, C*H*_3_); ^13^C NMR (CDCl_3_, 100 MHz): *δ* 150.9, 138.5, 137.1, 132.9, 124.5, 115.0, 114.6, 36.8, 21.7; yield, 76%.

(4-Methyl-1*H*-benzo[*d*]imidazol-2-yl)methanethiol (**9**)

^1^H NMR (DMSO-*d*_6_, 400 MHz): *δ* 7.36 (d, 1H, *J* = 8.0 Hz, 7-H), 7.08 (t, 1H, *J* = 8.0 Hz, 6-H), 6.98 (d, 1H, *J* = 8.0 Hz, 6-H), 4.24 (s, 2H, C*H*_2_), 2.52 (s, 3H, C*H*_3_); ^13^C NMR (DMSO-*d*_6_, 100 MHz): *δ* 150.5, 139.0, 138.4, 125.3, 122.8, 122.5, 112.6, 36.7, 17.2; yield, 80%.

(5,6-Dimethyl-1*H*-benzo[*d*]imidazol-2-yl)methanethiol (**10**)

^1^H NMR (DMSO-*d*_6_, 400 MHz): *δ* 7.30 (s, 2H, 4-H, 7-H), 4.11 (s, 2H, C*H*_2_), 2.29 (s, 6H, 2 × C*H*_3_); ^13^C NMR (DMSO-*d*_6_, 100 MHz): *δ* 150.0, 137.7, 130.8, 115.5, 36.4, 20.5; yield, 73%.

### 2.2. Materials for Biological Experiments

Pyrocatechol violet was purchased from the Tokyo Chemical Industry (Tokyo, Japan). Fetal bovine serum (FBS), Dulbecco Modified Eagle Medium (DMEM), and 2′,7′-dichlorodihydrofluorescein diacetate (DCFH-DA) were purchased from Thermo Fisher Scientific (Waltham, MA, USA). EZ-Cytox solution was purchased from DoGenBio (Seoul, Republic of Korea). Tricaine methanesulfonate, 3-isobutyl-1-methylxanthine (IBMX), 3-morpholinosydnonimine (SIN-1), α-melanocyte-stimulating hormone (α-MSH), 2,2′-azino-bis(3-ethylbenzothiazoline-6-sulfonic acid) (ABTS), Trolox, 2,2-diphenyl-1-picrylhydrazyl (DPPH), and mushroom TYR were purchased from Sigma-Aldrich Co. (St. Louis, MO, USA).

### 2.3. Activity Inhibition Assay Against Mushroom TYR [38,39]

A substrate solution (170 μL) containing l-dopa or l-tyrosine (345 μM) and sodium phosphate buffer (17.2 mM, pH 6.5) and a mushroom TYR aqueous solution (20 μL, 800 uints/mL) were subsequently added to each well of a 96-well plate containing kojic acid (10 μL) or 2-MMBI derivative (10 μL). Test samples (2-MMBIs **1**–**10** and kojic acid) in DMSO were treated with three to six different concentrations to obtain the IC_50_ values. After keeping the assay mixture in an incubator set to 37 °C for 15 min for l-dopa or for a half hour for l-tyrosine, the well optical density was measured at 475 nm using a VersaMax™ ELISA reader (Molecular Devices, Sunnyvale, CA, USA). Experiments were independently performed in triplicate.

### 2.4. Copper(II) Ion-Chelating Activity [40]

Copper(II) ion-chelating activity was measured using pyrocatechol violet reagent, a copper(II)-chelating agent. Copper(II) sulfate solution (10 μL, 1 mg/mL) was added to each well of a 96-well microplate containing acetic acid–sodium acetate buffer (280 μL, 50 mM [pH 6.0]), 2-MMBI derivative (**1**–**10**; 10 μL, 100 μM as a final concentration), and pyrocatechol violet (6 μL, 4 mM). Absorbance was measured at 632 nm using a VersaMax™ reader after the microplate was incubated for 20 min at 24 °C.
Cu(II) ion-chelating activity (%) = 100 × [(Abs_con_ − Abs_sam_)/Abs_con_]
where Abs_con_ and Abs_sam_ are the absorbances of the control and samples, respectively. Each experiment was independently performed in triplicate.

### 2.5. Mushroom TYR Activity Assay With and Without CuSO_4_ [41]

To determine whether copper(II) ions influence the TYR-inhibitory activity of 2-MMBI derivatives, the effects of these derivatives on mushroom TYR activity were measured with and without CuSO_4_. Briefly, mushroom TYR aqueous solution (20 μL, 1 unit/μL) was mixed with 2-MMBI derivatives (**1**, **3**, and **8**–**10**) (10 μL, final concentration: 20 μM) and with the substrate solution (170 μL), comprising 17.2 mM phosphate buffer (pH 6.5) and 345 μM l-tyrosine, in the presence or absence of CuSO_4_ (50 μL, final concentration: 50 μM) in each well of a 96-well plate. The optical density was measured at 475 nm using a VersaMax™ reader after incubation at 37 °C for 30 min. Each assay was carried out independently in triplicate.

### 2.6. Kinetic Studies on Mushroom TYR in the Presence of 2-MMBI Derivatives [42,43,44]

Kinetic experiments were performed to obtain Lineweaver–Burk plots for the analysis of mushroom TYR kinetics. Mushroom TYR solution (20 μL, 150 units/mL) was added to each well of a 96-well microplate containing 2-MMBI derivatives (10 μL; 0, 7.5, 15, and 30 μM for **1;** 0, 5, 10, and 20 μM for **3** and **8**; and 0, 12.5, 25, and 50 μM for **9**) and a substrate solution (170 μL) composed of sodium phosphate buffer (17.2 mM; pH 6.5) and l-dopa (various concentrations: 0.5, 1, 2, 4, 8, and 16 mM). The well optical density was measured at 475 nm every 1 min during the 15 min incubation period at 37 °C to calculate the change in the well optical density (ΔOD_475_/min). Lineweaver–Burk plots for each 2-MMBI derivative were obtained by plotting the value of 1/[substrate] against the value of (min/ΔOD_475_). Lineweaver–Burk plots were transformed into corresponding Dixon plots by plotting the value of (min/ΔOD_475_) against the value of the 2-MMBI derivative concentration.

### 2.7. Docking Simulation of 2-MMBI Derivatives and Mushroom TYR [29]

For in silico docking simulation, the 3D X-ray structure of *Agaricus bisporus* (https://www.rcsb.org/structure/2Y9X (accessed on 3 June 2024)) was used as the mushroom TYR enzyme. The 3D structures of the 2-MMBI derivatives **1**, **3**, **8**, and **9** were prepared using Chem3D Pro 12.0 (PerkinElmer Inc., Waltham, MA, USA, http://www.cambridgesoft.com/ (accessed on 3 June 2024)). Prior to the docking simulation, unnecessary B–H chains and water were deleted. After deleting the original ligand (tropolone) from the TYR structure, the 3D ligand structure was docked to the 3D TYR structure using AutoDock Vina (ver. 1.1.3) to determine the binding energies between TYR and the ligand. Information for pharmacophores was acquired using LigandScout 4.4 (InteLigand, https://ligandscout.software.informer.com, Vienna, Austria; accessed on 5 June 2024 and 12 July 2024). Prior to the docking simulation using the 2-MMBI derivatives, a redocking procedure with the crystallized ligand (tropolone) was performed to validate whether the ligand-free TYR protein could properly accommodate the crystallized ligand (Appendix A).

### 2.8. B16F10 Murine Melanoma Cell and HaCaT Keratinocyte Cell Culture

B16F10 and HaCaT cells purchased from the American Type Culture Collection (Manassas, VA, USA) were cultured in a solution containing DMEM, 10% heat-inactivated FBS, and 100 units/mL penicillin-streptomycin solution in an incubator set to the following conditions: 37 °C and 5% CO_2_.

### 2.9. B16F10 Cell Cytotoxicity Assay [45,46]

A 96-well plate containing 1 × 10^3^ B16F10 cells per well was incubated at 37 °C, with 5% CO_2_ for 22 h. 2-MMBI derivatives **1**–**10** at various concentrations (0, 2, 5, and 10 μM) were added to each well, and the plate was subsequently incubated at 37 °C and 5% CO_2_ for 72 h. The well optical density was measured at 450 nm using a VersaMax™ reader to determine cell viability after treatment with 10 μL EZ-Cytox solution for 2 h. Experiments were performed independently and repeated five times.

### 2.10. Melanin Content Measurement Assay [47]

A 6-well plate containing 5 × 10^3^ B16F10 cells per well was incubated at 37 °C and 5% CO_2_ for 1 day. 2-MMBI derivative (**3**, **7**, **8**, and **10**; final concentrations: 2, 5, and 10 μM) or kojic acid (10 μM, positive material) was added to each well. After 1 h, 1 μM α-MSH and 200 μM IBMX were added to each well, and the plate was subsequently incubated at 37 °C and 5% CO_2_ for 72 h. The cells were exposed to 100 μM 1N-NaOH solution and cultured at 60 °C for 1 h after washing with PBS. The lysates were transferred to each well of a 96-well plate, and the optical density at 405 nm was recorded using a VersaMax™ reader. In the preliminary experiments, all test samples (2-MMBI derivatives and kojic acid) were administered at concentrations of 10 μM. Each assay was independently performed three times.

### 2.11. TYR Activity Assay in B16F10 Cells [45]

A 6-well plate containing 5 × 10^3^ B16F10 cells per well was incubated (37 °C and 5% CO_2_) for 1 day. A 2-MMBI derivative (final concentrations: 2, 5, and 10 μM) or kojic acid (10 μM, positive material) was added to each well. Stimulators (1 μM α-MSH and 200 μM IBMX) were added after 1 h, and the plate was incubated (37 °C and 5% CO_2_) for 72 h. After washing with PBS, 100 μL lysis buffer consisting of 1% Triton X-100, 1 mM phenylmethylsulfonyl fluoride, and 50 mM sodium phosphate buffer (pH 6.5) in a volume ratio of 1:1:18 was added to each well. Following an incubation period of 30 min at −80 °C, the cell lysates were centrifuged at 12,000× *g* for 10 min at 4 °C to obtain the supernatants. The supernatants (80 μL) were mixed with 10 mM l-dopa (20 μL) in each well of a 96-well plate, and then the well optical density was recorded at 475 nm every 10 min for 60 min using a VersaMax™ reader. Each assay was independently performed three times.

### 2.12. In Situ B16F10 Cellular TYR Activity Assay [48]

The in situ cellular TYR activities of the 2-MMBI derivatives were investigated using B16F10 cells. B16F10 cells (2 × 10^3^/well) were inoculated in each well of a 24-well microplate and cultured for 1 day in an incubator (37 °C and 5% CO_2_). Cells were pretreated with test samples (2-MMBI derivatives **7** and **10** [0, 2, 5, and 10 μM] and kojic acid [10 μM; positive material]) for 1 h. Subsequently, stimulators (α-MSH [1 μM] and IBMX [200 μM]) were added to each well. Following an incubation period of 72 h at 37 °C and 5% CO_2_, the cells were fixed, washed, and permeabilized using 4% paraformaldehyde, PBS, and 0.1% Triton X-100, respectively. After rinsing with PBS, cells were exposed to l-DOPA (2 mM, 500 μL) at 37 °C for 2 h. Stained photographs were captured using a camera attached to a microscope (Motic, Hong Kong, China). Each assay was independently performed two times.

### 2.13. Melanogenesis Assay Using Zebrafish Embryos [30,49]

To assess depigmentation in vivo, wild-type zebrafish (*Danio rerio*) embryos were acquired free of charge from the Zebrafish Center for Disease Modeling (ZCDM) located at Chungnam National University in Daejeon, Republic of Korea. The zebrafish were raised in tanks maintained at 28 °C in the ZCDM. The tanks were oxygenated, and the tank lights were turned off for 10 h and on for 14 h. The zebrafish embryos used for in vivo depigmentation experiments were obtained through natural mating. The E3 solution for zebrafish embryos was prepared by dissolving 19.9 mg magnesium sulfate, 6.4 mg potassium chloride, 18.3 mg calcium chloride, and 146.1 mg sodium chloride in 500 mL distilled water. E3-MB (methylene blue) solution was prepared by adding MB to the E3 solution to create a 0.001% MB solution. The acquired zebrafish embryos were transferred to a culture dish containing the E3-MB solution. The culture dish was incubated at 28 °C until zebrafish embryos were used. Since zebrafish embryos are sensitive and can be easily killed by the chorion removal process 20 h post-fertilization (20 hpf), the chorion was removed at 24 hpf through treatment with pronase (Sigma-Aldrich, St. Louis, MO, USA). Five dechorionated zebrafish embryos were transferred to each well of 48-well microplates containing 250 μL E3 solution. At 28 hpf, each well of 48-well microplate was treated with test samples (2-MMBI derivatives **1**–**10** [0.03 and 0.1 mM] and kojic acid [20 mM]), and the microplates were incubated for 48 h at 28 °C. Zebrafish larvae were anesthetized with tricaine at 76 hpf and placed on a 1% methylcellulose block to determine the degree of zebrafish larva depigmentation. Dorsal and lateral images of the zebrafish larvae were acquired using an SMZ745T microscope (Nikon, Tokyo, Japan). The pigmented areas of the dorsal and lateral images were obtained using CS analyzer 3.0 image analysis software (ATTO, Tokyo, Japan). Each assay (n = 7) was independently performed two times.

### 2.14. Cell Viability Assay in HaCaT Cells [50]

A 96-well plate containing 1 × 10^5^ HaCaT cells per well was incubated at 37 °C and 5% CO_2_ for 20 h. 2-MMBI derivatives at various concentrations (0, 2, 5, 10, and 20 μM) were added to each well, and the plate was incubated at 37 °C, 5% CO_2_ for 24 h. After treatment with 10 μL EZ-Cytox solution for 2 h, the well optical density was measured at 450 nm using a VersaMax™ reader to determine cell viability. Each assay was independently performed five times.

### 2.15. ABTS^●+^ Scavenging Assay [51,52]

A 2.45 mM K_2_S_2_O_8_ aqueous solution was mixed with a 7-mM ABTS aqueous solution in a 1:1 volume ratio. The mixture was left at 20 °C in the dark for 19 h. To adjust the adequate absorbance (0.70 ± 0.01) at 732 nm, the mixture was diluted with methanol. The diluted ABTS^●+^ mixture (90 μL) in each well of a 96-well plate was mixed with a test sample (2-MMBI derivatives **1**–**10** and Trolox [positive material]; 10 μL in EtOH/DMSO [9/1 (*v*/*v*)] solution). All test samples were tested at a final concentration of 100 μM. Prior to the measurement of well optical density at 732 nm at 1-min intervals for 10 min using a VersaMax™ reader, the test sample-ABTS^●+^ mixture was stored for 2 min in the dark. Each experiment was conducted independently in triplicate.

### 2.16. DPPH Radical Removal Assay [53]

An aliquot (20-μL DMSO solution) of a test sample (5 mM 2-MMBI derivatives **1**–**10** and 5 mM vitamin C [positive material]) was added to each well of a 96-well microplate containing 180 μL DPPH methanol solution (0.2 mM). After the mixture was left in the dark at 22 °C for 30 min, the optical density at 517 nm was recorded using a VersaMax™ reader. Each experiment was conducted independently in triplicate.

### 2.17. Reactive Oxygen Species Removal Activity Assay [54,55]

To assess reactive oxygen species (ROS) removal activity, a DCFH solution was prepared by mixing 50 μL DCFH-DA (1.25 mM), 50 μL esterase (0.6 unit/μL), and 4.9 mL phosphate buffer (50 mM) and maintained in the dark for 30 min. The SIN-1 solution (10 μL) was added to each well of a 96-black well that contained the test sample (10 μL; 2-MMBI derivatives **1**–**10** and Trolox [positive material]) and phosphate buffer (180 μL). 2-MMBI derivatives, Trolox, and SIN-1 were exposed at final concentrations of 40, 40, and 10 μM, respectively. The prepared DCFH solution (50 μL) was added to each well after 10 min. The fluorescence intensity of 2′,7′-dichlorofluorescein in the wells was measured at 535 nm using a microplate reader (Berthold Advances GmbH & Co., Bad Wildbad, Germany) with an excitation wavelength of 485 nm. Each experiment was conducted independently four times.

### 2.18. Statistical Analysis

Experimental data were represented as the mean ± standard error of the mean. GraphPad Prism (La Jolla, CA, USA) was utilized for statistical analyses: one-way analysis of variance followed by the Bonferroni post hoc test. Statistical significance was set to a *p*-value < 0.05.

## 3. Results and Discussion

### 3.1. Synthesis of 2-MMBI Derivatives **1**–**10**

To synthesize the target compounds (2-MMBI derivatives **1**–**10**), we employed a condensation reaction between 1,2-phenylenediamine and carboxylic acid under strongly acidic conditions, which is a widely used method for the synthesis of benzo[*d*]imidazole rings [56,57]. Among the different 1,2-phenylenediamines, 4-trifluoromethyl-1,2-phenylenediamine (**11**) was synthesized from 4-trifluoromethyl-2-nitroaniline through Bechamp reduction using iron and ammonium chloride, as shown in Figure 1. Refluxing an appropriate 1,2-phenylenediamine and thioglycolic acid in 4N-HCl achieved corresponding benzo[*d*]imidazole derivatives with yields ranging from 52% to 98%.

As the fused benzo[*d*]imidazole rings were formed through the condensation reaction, the proton peaks of the benzene ring in the ^1^H-NMR spectra of compounds **1**–**10** appeared to shift downfield (≥6.90 ppm) due to the anisotropic effect of the imidazole ring. In ^13^C-NMR spectra, carbon peaks were observed in the range of 156.8–150.0 ppm. This is the range in which the carbon peak at position 2 of the benzo[*d*]imidazole appeared, further proving that a benzo[*d*]imidazole ring was formed.

### 3.2. Inhibitory Effect of 2-MMBI Derivatives on Mushroom TYR Activity

The TYR inhibitory abilities of 2-MMBI derivatives, which were designed and synthesized as TYR inhibitors, were evaluated using commercially available mushroom TYR. Kojic acid was used as a positive control to compare the TYR inhibitory activities.

First, the mushroom TYR inhibitory activities of the 2-MMBI derivatives were investigated in the presence of l-tyrosine as a TYR substrate (Table 1). Kojic acid strongly inhibited TYR activity, with an IC_50_ value of 17.87 μM. Of the synthesized derivatives **1**–**10**, seven 2-MMBI derivatives demonstrated a more potent TYR inhibitory activity than that of kojic acid, which is used as a skin-whitening agent in some countries. Derivative **1** (IC_50_ value: 4.05), which has no substitution effect on the benzene of the benzo[*d*]imidazole ring, inhibited TYR activity 4.4 times more strongly than kojic acid. The substitution of an electron-withdrawing group (EWG) on the benzo[*d*]imidazole ring decreased the TYR inhibitory activity. Derivatives **2** and **4**, which were substituted with 5-Cl and 5-F on the benzo[*d*]imidazole ring of **1**, respectively, revealed reduced TYR inhibitory activities, with IC_50_ values of 15.36 and 16.29 μM, respectively. However, their TYR inhibitory activities were still slightly stronger than that of kojic acid. Derivatives **5** and **6**, in which the benzo[*d*]imidazole ring of **1** was substituted with 5-CN and 5-NO_2_, respectively, showed slightly greater decreases in their TYR inhibitory activities, with IC_50_ values of 21.89 and 20.91 μM, respectively. Derivative **7**, in which the benzo[*d*]imidazole ring of **1** was substituted with strongly electron-withdrawing and bulky 5-CF_3_, had strongly reduced TYR inhibitory activity, and its IC_50_ value (52.37 μM) was 13 times higher than that of **1**. Conversely, the substitution of an electron-donating group (EDG) on the benzo[*d*]imidazole ring slightly decreased or increased the TYR inhibitory activity. Derivative **3**, substituted with 5-OMe on the benzo[*d*]imidazole ring of **1**, showed a slightly decreased TYR inhibitory activity with an IC_50_ value of 6.15 μM, while derivative **8**, substituted with 5-CH_3_ on the benzo[*d*]imidazole ring of **1**, exhibited slightly increased TYR inhibitory activity with an IC_50_ value of 2.90 μM. The TYR inhibitory effect of derivative **8** was 6 times higher than that of kojic acid. As in derivative **8**, the insertion of a CH_3_ group at the fifth position of the benzo[*d*]imidazole ring of **1** enhanced the TYR inhibitory efficacy. However, the insertion of the same group into the fourth position of the benzo[*d*]imidazole ring of **1** decreased the TYR inhibitory efficacy (IC_50_ value of derivative **9**: 11.86 μM). However, the inhibitory potency of **9** against the mushroom TYR was superior to that of kojic acid. Inserting a CH_3_ group into positions 5 and 6 on the benzo[*d*]imidazole ring of **1** had little effect on the TYR inhibitory activity (IC_50_ value of derivative **10**: 5.66 μM).

Second, in the presence of l-dopa as a TYR substrate, the mushroom TYR inhibitory activities of 2-MMBI derivatives were examined (Table 1). Kojic acid exhibited strong TYR inhibitory activity with an IC_50_ value of 23.53 μM, which was slightly higher than that when l-tyrosine was used. Furthermore, for most derivatives, their IC_50_ values in the presence of l-dopa were higher than those in the presence of l-tyrosine. Unlike in the presence of l-tyrosine, of the ten derivatives, only three 2-MMBI derivatives (**1**, **3**, and **8**) demonstrated more potent TYR inhibitory activity than kojic acid. The relationships between the substituents and TYR inhibitory activities were similar to those observed when l-tyrosine was used as the TYR substrate. Derivative **1** (IC_50_ value: 13.78 μM), with no substitution on the benzene of the benzo[*d*]imidazole ring, demonstrated more potent TYR activity than kojic acid. The insertion of EWGs into the benzene ring of the benzo[*d*]imidazole ring reduced the TYR inhibitory activities as follows: **2** with 5-Cl, **4** with 5-F, **5** with 5-CN, **6** with 5-NO_2_, and **7** with 5-CF_3_ had IC_50_ values of 51.56, 35.26, 73.56, 53.75, and 46.07 μM, respectively. Moreover, the effect of the EDG substitution on TYR inhibitory activity varied depending on the substitution position: the substitution of EDGs, such as OMe and CH_3_, at the fifth position of the benzo[*d*]imidazole ring slightly increased the TYR inhibition (IC_50_ values of **3** with 5-OMe and **8** with 5-CH_3_ were 9.77 and 10.64 μM, respectively), whereas the substitution of CH_3_ at the fourth position of the benzo[*d*]imidazole ring resulted in a decrease in TYR activity inhibition (IC_50_: **9**: 28.97 μM). CH_3_ insertion at positions 5 and 6 decreased the TYR inhibitory activity (IC_50_: **10**: 54.26 μM).

These results suggest that the three 2-MMBI derivatives, **1**, **3**, and **8**, which each exhibited a stronger mushroom TYR inhibition potency than that of kojic acid on both substrates (l-tyrosine and l-dopa), could serve as promising anti-browning agents to prevent browning and improve the quality of certain crops, such as fruits and vegetables, during storage.

### 3.3. Copper Ion Chelation Ability of 2-MMBI Derivatives

The ability of the 2-MMBI derivatives to chelate copper ions was tested using pyrocatechol violet, a copper-ion-chelating reagent. The CuSO_4_ solution was mixed with a buffer solution (pH 6.0), 2-MMBI derivative solution, and pyrocatechol violet solution. After an incubation period of 20 min, the absorbance of the mixture was recorded at 632 nm to determine the copper ion chelating activity. Kojic acid and PTU, which chelate copper ions, were used as positive controls. All the test samples were used at a final concentration of 100 μM.

The results of copper chelation are shown in Figure 2. PTU and kojic acid showed 37% and 27% copper chelating activities, respectively. Most of the 2-MMBI derivatives exhibited approximately 20% copper chelating activities. Of the 2-MMBI derivatives, the derivative **6** with a 5-NO_2_ substitution showed the highest copper-chelating potential with a 35% copper chelating activity, which was comparable to that of PTU.

The copper-chelating and mushroom TYR inhibitory activities of the derivatives were compared. Derivative **6** revealed the highest copper-chelating activity but lower mushroom TYR inhibitory activity than most other derivatives. These results suggest that in addition to the copper-chelating abilities of the derivatives, the chemical interactions between the chemical structures of the derivatives and the amino acid residues of the TYR active sites are also significantly involved in TYR inhibitory activities. The insertion of bulky substituents, such as nitro and trifluoromethyl, at the fifth position of the benzo[*d*]imidazole ring, as in derivatives **6** and **7**, appears to decrease the TYR inhibitory activity due to steric hindrance.

Recently, we reported the TYR inhibitory activities of 2-mercapto-*N*-arylacetamide analogs. Although these analogs demonstrated potent anti-TYR activities, they did not chelate copper ions. We hypothesized that the unshared electron pair of the amide nitrogen formed a resonance structure with the neighboring carbonyl group, which lowered the electron density of the unshared electron pair of the amide nitrogen, making chelation with copper ions difficult (Figure 3A). In the case of 2-MMBIs, the unshared electron pair of the N^1^ atom was used to form the aromaticity of the imidazole ring; therefore, it was impossible for the N^1^ atom to participate in chelation with copper ions (Figure 3B). However, the unshared electron pair on the N^3^ atom did not contribute to the aromaticity of the imidazole ring. Instead, it exhibited a greater electron density due to the resonance structure formed between the N^3^ and N^1^ atoms (Figure 3B). Therefore, unlike 2-mercapto-*N*-arylacetamides, the N^3^ atom of 2-MMBIs could easily chelate copper ions with the sulfur of the 2-mercapto group.

### 3.4. Decreasing Mushroom TYR Activity in the Presence of CuSO_4_

We explored whether the presence of CuSO_4_ affected the TYR inhibitory activities of the 2-MMBI derivatives. Mushroom TYR and l-tyrosine were used as the test enzymes and substrates, respectively. Since derivatives **1**, **3**, and **8**–**10** showed potent TYR inhibitory activities in the presence of l-tyrosine, these derivatives were used as test samples and treated at a single concentration of 20 μM.

As shown in Figure 4, when no CuSO_4_ was added externally, derivatives **1**, **3**, **8**, and **9** inhibited TYR activities by 86–89%, and derivative **10** inhibited TYR activity by 60%. However, when CuSO_4_ was added externally, the TYR inhibitory activities of derivatives **1**, **3**, **8**, **9**, and **10** were all reduced to 76, 34, 63, 54, and 1%, respectively. This phenomenon occurred because the 2-MMBI derivatives were chelated by externally added CuSO_4_, which decreased the degree of chelation of the 2-MMBI derivative with TYR copper ions, thereby reducing the TYR inhibitory ability of the 2-MMBI derivatives.

### 3.5. Mechanism Studies on TYR Inhibition of 2-MMBI Derivatives

Because the 2-MMBI derivatives **1**, **3**, **8**, and **9** exhibited potent mushroom TYR inhibitory efficacy, their inhibitory modes of action were examined. To determine the inhibitory mechanisms of 2-MMBI derivatives against mushroom TYR, the rates of dopachrome formation at four different concentrations of 2-MMBI derivatives were measured in the presence of five or six l-dopa concentrations. A Lineweaver–Burk plot for each derivative was obtained based on the rate of dopachrome formation at various concentrations (Figure 5). Derivatives **1** and **3** each produced a Lineweaver–Burk plot consisting of four straight lines with different slopes intersecting the y-axis, whereas derivatives **8** and **9** each produced a Lineweaver–Burk plot consisting of four straight lines with different slopes that intersected the second quadrant. These results indicate that **1** and **3** are competitive TYR inhibitors that compete with substrates at the TYR active site, whereas **8** and **9** are mixed TYR inhibitors that can bind to both the TYR active and allosteric sites.

To assess the potency of the inhibitors, each Lineweaver–Burk plot for derivatives **1**, **3**, **8**, and **9** was converted into the corresponding Dixon plots (Figure 6) to determine the inhibition constant (K_i_). The straight lines in the Dixon plots converge at a point above the x-axis, with the negative x-value of the point representing K_i_, the concentration required to achieve half the maximum inhibition. The K_i_ values of **1**, **3**, **8**, and **9** were 14.54, 7.92, 12.96, and 29.18 μM, respectively, which suggests that derivative **3** binds most strongly to the mushroom TYR.

### 3.6. Docking Simulation of Mushroom TYR and 2-MMBI Derivatives

As 2-MMBI derivatives **1**, **3**, **8**, and **9** effectively inhibited mushroom TYR activity, in silico docking simulations between these ligands (**1**, **3**, **8**, and **9**) and mushroom TYR were performed using AutoDock Vina to examine their binding affinities and chemical interactions with each other. The Protein Data Bank (PDB) was used to acquire the TYR X-ray structure. The PDB ID 2Y9X for *A. bisporus* was used for the docking simulation. After deleting the original ligand (tropolone), derivatives **1**, **3**, **8**, and **9** or kojic acid (positive control) were docked into the tropolone-bound TYR active site.

The docking results are presented in Figure 7. Kojic acid was involved in two interactions: a hydrogen bonding of the 2-hydroxymethyl with Asn260 and pi–pi stacking of the pyranone ring with His263. These interactions provided kojic acid with a binding energy of −5.4 kcal/mol. Derivatives **1** and **3** were involved in hydrophobic interactions with the same two amino acids (Ala286 and Val283), giving these derivatives binding energies of −5.6 and −6.0 kcal/mol, respectively. The benzene rings of derivatives **8** and **9** with 5-methyl and 4-methyl groups, respectively, were also involved in hydrophobic interactions with Val283. The 5-methyl group of compound **8** hydrophobically interacted with three amino acids, Phe292, Ala286, and Val283, whereas the 4-methyl group of compound **9** hydrophobically interacted with two amino acids, Phe264 and Val283. These interactions gave **8** and **9** binding energies of −5.5 and −5.9 kcal/mol. Since Val283 was involved in the interactions of all derivatives and Ala286 was involved in the interactions of derivatives **1**, **3**, and **8**, these two amino acid residues appeared to play an important role in binding to the TYR active site. The docking simulation results suggest that the 2-MMBI derivatives bind to the TYR active site similarly or more strongly than kojic acid.

Because the kinetic study results suggested that derivatives **8** and **9** were mixed inhibitors, we investigated whether **8** and **9** could bind to the allosteric site of mushroom TYR using AutoDock Vina. Derivatives **8** and **9** were docked into the allosteric site of TYR, and the results are shown in Figure 8.

Derivatives **8** and **9** occupied the same allosteric sites (Figure 8A,B). According to the chemical interaction results, the mercapto substituent of **8** formed hydrogen bonds with Glu356 and Asp312 as hydrogen bond donors and acceptors, respectively (Figure 8C,D). Furthermore, the 5-methyl and benzene rings hydrophobically interact with Phe368, Thr308, and Trp358. Derivative **9** also formed a hydrogen bond with Glu356 as a hydrogen bond donor, Trp358 hydrophobically interacted with 4-methyl, and the benzene ring, and Thr308 hydrophobically interacted with the benzene ring. These interactions provided derivatives **8** and **9** binding energies of −5.4 and −6.0 kcal/mol, respectively. These derivatives (**8** and **9**) showed similar binding energies at the active and allosteric sites, which supported the kinetic study results that these derivatives acted as mixed TYR inhibitors.

### 3.7. B16F10 Cell Viability in the Presence of 2-MMBI Derivatives

Prior to the B16F10 cell-based experiments, the effects of 2-MMBI derivatives on the viability of B16F10 cells were examined. 2-MMBI derivatives were added to B16F10 cells cultured in DMEM containing 5% FBS, and the cell viability was determined using an EZ-Cytox assay after a 72-h incubation period. 2-MMBI derivatives were tested at three concentrations: 2, 5, and 10 μM.

Figure 9 shows the B16F10 cell viability results for the 2-MMBI derivatives. None of the derivatives demonstrated significant cytotoxicity against B16F10 cells at any of the tested concentrations. Thus, B16F10 cell-based experiments were performed at concentrations ≤ 10 μM, which did not show cytotoxicity to B16F10 cells.

### 3.8. Effects of 2-MMBI Derivatives on Melanogenesis in B16F10 Cells

We investigated whether the 2-MMBI derivatives could inhibit melanin production in B16F10 murine cells. Due to the structural differences between mushroom TYR, which exists as a tetramer, and mammalian TYR, which exists as a glycosylated monomer, 2-MMBI derivatives that exhibit potent mushroom TYR inhibitory activity may not inhibit mammalian TYR. Thus, the inhibitory effects of 2-MMBI derivatives on melanogenesis were evaluated at a single concentration of 10 μM using B16F10 cells, which led to the selection of 2-MMBI derivatives for further detailed cell experiments. B16F10 cells were first treated with 10 μM 2-MMBI derivatives for 1 h and then treated with stimulators consisting of 1 μM α-MSH and 200 μM IBMX, and the inhibitory effects of the derivatives on melanin production were determined after 72 h.

The antimelanogenic results are shown in Figure 10. Exposure to stimulators increased melanin content; however, treatment with 10 μM of the 2-MMBI derivatives or kojic acid (positive control) decreased the stimulator-induced melanin levels. Particularly, derivatives **3**, **7**, **8**, and **10** demonstrated stronger inhibitory effects on melanin production than the other derivatives and inhibited melanin production more strongly than kojic acid. Therefore, these four derivatives were selected for further investigation of their inhibitory effects on cellular TYR activity and melanin production. Notably, due to structural differences between mushroom TYR and B16F10 cell TYR, derivative **1** exhibited strong mushroom TYR inhibitory activity but a relatively low melanin inhibitory effect in B16F10 cells, whereas **7** exhibited relatively low mushroom TYR inhibitory activity but the strongest melanin inhibitory effect in B16F10 cells.

The inhibitory effects of the selected derivatives **3**, **7**, **8**, and **10** on melanin production in B16F10 cells were further investigated at concentrations of 2, 5, and 10 μM, and kojic acid (10 μM) was utilized for comparing inhibitory activity. B16F10 cells were first exposed to test samples for 1 h and then exposed to stimulators (200 μM IBMX and 1 μM α-MSH). The melanin production was measured after 72 h.

Treatment with the stimulators significantly increased the melanin content by 4.4-fold, but exposure to kojic acid lowered the melanin content by 3.8-fold (Figure 11). Additionally, four derivatives (**3**, **7**, **8**, and **10**) significantly decreased the melanin levels increased by the stimulators in a concentration-dependent manner. When compared at the same concentration (10 μM), all derivatives exhibited much more potent melanin inhibition abilities than kojic acid. Derivatives **7** and **10** inhibited melanin production more strongly than derivatives **3** and **8**. In particular, derivative **7** potently reduced the stimulator-induced increase in melanin content to the levels observed in the control group.

### 3.9. Effects of 2-MMBI Derivatives on Cellular TYR Activity Inhibition in B16F10 Cells

Because the 2-MMBI derivatives **3**, **7**, **8**, and **10** potently inhibited melanogenesis in B16F10 cells, we investigated whether the inhibition of melanogenesis occurred due to their TYR inhibitory abilities.

As in the melanin content experiment with B16F10 cells, the cells were pretreated for 1 h with test samples (**3**, **7**, **8**, and **10**: 2, 5, and 10 μM or kojic acid [positive material; 10 μM]) and subsequently treated with stimulators (200 μM IBMX and 1 μM α-MSH). After a 72-h incubation period, the B16F10 cellular TYR activity of the test samples was determined.

Exposure to these stimulators significantly increased cellular TYR activity by 3.9-fold (Figure 12). Treatment with kojic acid reduced the stimulator-induced increase in cellular TYR activity to 2.9-fold. Derivatives **3**, **7**, **8**, and **10** significantly reduced stimulator-enhanced cellular TYR activity in a dose-dependent manner. As observed in the melanin content experiments, derivatives **7** and **10** more strongly inhibited cellular TYR than derivatives **3** and **8**. Overall, the inhibition of cellular TYR and melanin production in B16F10 cells was similar. This finding supports the hypothesis that the inhibitory effects of 2-MMBI derivatives on melanin formation occur due to their abilities to inhibit TYR. Based on the melanin inhibition results of the 2-MMBI derivatives, **7** was thought to have anti-melanogenesis mechanisms in addition to its TYR inhibitory activity. Therefore, the antioxidant activities of the 2-MMBI derivatives, such as their ROS-scavenging abilities, were evaluated in further mechanistic studies.

### 3.10. In Situ B16F10 Cell TYR Activity in the Presence of 2-MMBI Derivatives

To investigate the effects of the 2-MMBI derivatives on TYR activity in B16F10 cells, B16F10 cells were stained with l-dopa. This method uses the principle that when cellular TYR activity increases, cells produce more melanin using the substrate l-dopa, causing the cells to become darker when stained with melanin. 2-MMBI derivatives **7** and **10** were treated at concentrations of 2, 5, and 10 μM, and kojic acid, a positive control, was treated at 10 μM. An hour after treatment with test samples (**7**, **10**, and kojic acid), α-MSH (1 μM) and IBMX (200 μM) were treated. After 72 h, excess l-dopa was added, and the cells were incubated for 2 h to allow melanin production.

The results of the in situ B16F10 cell TYR activity assays are shown in Figure 13. Exposure to α-MSH and IBMX greatly enhanced cell staining; however, treatment with kojic acid somewhat reduced the increased cell staining induced by α-MSH and IBMX treatment. Exposure to derivatives **7** and **10** also reduced the increase in cell staining induced by α-MSH and IBMX in a concentration-dependent manner. When compared at the same concentration (10 μM), **7** and **10** inhibited in situ B16F10 cell TYR activity much more potently than kojic acid, with **10** showing a slightly better inhibitory effect than **7**.

### 3.11. Inhibitory Effect of 2-MMBI Derivatives on Pigmentation in Zebrafish Larvae

Zebrafish, mushrooms, and B16F10 cellular TYRs share the same function; however, they exhibit structural differences between themselves due to their varying origins. Consequently, compounds that do not inhibit one type of TYR may exhibit inhibitory activity against another type of TYR. Therefore, the effect on zebrafish depigmentation was examined for all 2-MMBI derivatives.

To determine the effect of the 2-MMBI derivatives on zebrafish larval pigmentation, zebrafish embryos were acquired from the Zebrafish Center for Disease Modeling (ZCDM) at Chungnam National University (Daejeon, Republic of Korea). The obtained zebrafish embryos were incubated at 28 °C before use, and at 24 hpf, the chorion of each zebrafish embryo was removed using protease (Figure 14A). After 4 h, zebrafish embryos were exposed to 0.03 and 0.1 mM of 2-MMBI derivatives or 20 mM kojic acid (a positive material) for 48 h. The degree of depigmentation in the zebrafish larvae was photographed from the back and sides.

Depigmentation results of zebrafish larvae are displayed in Figure 14B,C. Compared to the control, exposure to kojic acid significantly reduced melanin pigments (Figure 14B). The results of the 2-MMBI derivatives varied depending on the derivative. No depigmentation results were obtained for derivatives **4** and **7** because all zebrafish larvae died at the tested concentrations; this may be attributed to toxicity. Additionally, zebrafish larvae treated with 0.1 mM derivative **2** also died. Conversely, depigmentation results were obtained for the remaining derivatives without dead zebrafish larvae. Three derivatives, **2**, **3**, and **8**, exhibited much stronger depigmentation efficacy than kojic acid, even at a concentration of 0.03 mM, which is 660 times lower than the concentration of kojic acid. At 0.1 mM, which was 200 times lower than the concentration of kojic acid, derivatives **1**, **3**, and **8**–**10** showed much more potent inhibition of melanin production in zebrafish larvae than kojic acid. Conversely, derivatives **5** and **6** did not exhibit significant depigmentation. Five derivatives, **1**, **3**, and **8**–**10**, showed concentration-dependent results, whereas derivatives **5** and **6** did not appear to show concentration-dependent results. Therefore, to further analyze the pigmentation areas, we used a densitometer (CS analyzer) to determine the densities of the pigmentation images (Figure 14C). Pigmentation areas were measured at two locations, the dorsal and lateral views, and similar results were obtained. Both kojic acid and 2-MMBI derivatives significantly reduced pigmentation. In particular, the four derivatives **1**, **3**, and **8**–**10** reduced the pigmentation area more strongly than kojic acid in a concentration-dependent manner, even at concentrations hundreds of times lower than that of kojic acid. The IC_50_ values of 2-MMBI derivatives obtained using mushroom tyrosinase did not completely correlate with the melanogenesis or tyrosinase inhibition in cells or in vivo, probably due to structural differences between tyrosinases across species.

### 3.12. Cell Viability of 2-MMBI Derivatives in HaCaT Cells

For 2-MMBI derivatives to be effective as skin-whitening agents, they should not be cytotoxic to skin epidermal cells. Keratinocytes are the major cells in the epidermis layer. Thus, the cytotoxicity levels of the 2-MMBI derivatives **1**–**10** were examined in HaCaT (keratinocytes) cells. All derivatives were exposed to HaCaT cells at concentrations of 0, 2, 5, 10, and 20 μM for 24 h.

Figure 15 shows the viabilities of HaCaT cells in the presence of derivatives **1**–**10**. None of the derivatives exhibited cytotoxic effects in HaCaT cells. Altogether, the 2-MMBI derivatives were found to be suitable for skin applications, as they did not exhibit cytotoxic effects on melanocytes (B16F10) or keratinocytes (HaCaT), which are the major cells of the epidermis.

### 3.13. Antioxidant Activities of 2-MMBI Derivatives

Since it has been reported that antioxidant capacity is closely related to the regulation of melanin biosynthesis [58,59,60], the antioxidant activities of 2-MMBI derivatives **1**–**10** against 2,2′-azino-bis(3-ethylbenzothiazoline-6-sulfonic acid) radical cation (ABTS^●+^), 2,2-diphenyl-1-picrylhydrazyl (DPPH) radicals, and ROS were evaluated.

ABTS^●+^ was generated by mixing potassium persulfate and ABTS. The test samples (2-MMBI derivatives and Trolox [positive material]) were each treated at a concentration of 100 μM.

Trolox inhibited ABTS^●+^ by 99%, while only three 2-MMBI derivatives (**1**, **3**, and **10**) showed moderate ABTS^●+^ inhibition activities of 24–30% (Figure 16A). The remaining derivatives exhibited weak ABTS^●+^ inhibition activities lower than 15%.

The scavenging activities of the 2-MMBI derivatives against DPPH radicals were evaluated using l-ascorbic acid as a positive control. The DPPH radical scavenging activities of the derivatives were determined 30 min after mixing the DPPH and 2-MMBI derivative solutions in the dark. The test samples were treated at 500 μM.

L-Ascorbic acid revealed a strong DPPH radical scavenging activity (97% scavenging) (Figure 16B). Three 2-MMBI derivatives exhibited moderate DPPH radical scavenging activities of 43–56%, and these were the same derivatives that exhibited moderate ABTS^●+^ inhibition capacities. The remaining derivatives showed DPPH radical scavenging activities lower than 27%.

The scavenging activities of 2-MMBI derivatives on ROS were evaluated using Trolox as a positive control. SIN-1 (10 μM) was used as a ROS generator, and all test samples (**1**–**10** and Trolox) were treated at 40 μM.

As shown in Figure 16C, treatment with SIN-1 greatly increased in vitro ROS levels, but exposure to the test samples significantly decreased the SIN-1-increased in vitro ROS levels. Trolox exhibited the highest ROS scavenging activity. Among the derivatives, derivatives **2**, **6**, and **7** exhibited stronger ROS scavenging efficacy than the other derivatives.

In the B16F10 cell-based experiments, derivatives **7** and **10** exhibited similar cellular TYR inhibitory activities; however, **7** inhibited melanin production more strongly than **10** (Figure 11 and Figure 12). These results may arise partly due to the differences in the ROS scavenging activities of derivatives **7** and **10**.

## 4. Conclusions

Seven of the 2-MMBI derivatives **1**–**10**, which were synthesized as TYR chelators, exhibited stronger mushroom TYR inhibitory activities than kojic acid. These derivatives could chelate copper ions. Furthermore, kinetic studies of mushroom TYR indicated that among the four derivatives tested, two were competitive inhibitors, and two were mixed-type inhibitors, which was supported by the docking results. No 2-MMBI derivatives **1**–**10** were cytotoxic to melanocytes or keratinocytes at the concentrations tested. In B16F10 cell-based experiments, all derivatives significantly inhibited melanin production with a potency similar to or much stronger than that of kojic acid. Derivatives **3**, **7**, **8**, and **10** showed excellent efficacy in melanin reduction in a concentration-dependent manner and inhibited B16F10 cellular TYR activity, which was similar to the melanin content. Several derivatives exhibited extremely strong depigmentation effects on zebrafish larvae. Additionally, 2-MMBI derivatives showed moderate ROS-scavenging activities. These results suggested that 2-MMBI derivatives were versatile TYR inhibitors that could inhibit TYRs of various origins and have diverse applications in inhibiting melanogenesis.

## Data Availability

Data is contained within the article or Appendix A.

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
