# Peer review of "Design, Synthesis, and Anti-Melanogenic Activity of 2-Mercaptomethylbenzo[*d*]imidazole Derivatives Serving as Tyrosinase Inhibitors: An In Silico, In Vitro, and In Vivo Exploration"

_antioxidants, 2024, doi:10.3390/antiox13101248_

Round 1
Reviewer 1 Report
The paper present a clear workflow for developing TYR inhibitors using different approaches (in silico in vitro and in vivo). The paper is well organized and it is quite interesting, demonstrating the potential of the developed compounds as antioxidant agents. The manuscript deserves publication in Antioxidants after addressing the points reported below:
-purity of compounds should be reported
-number of independent experiments should be clearly stated for each test performed in the materials and methods
-docking protocol was not validated, there is not the redocking procedure with the crystallized ligand, to assess the capability of docking protocol to correctly accommodate the crystallized ligand, calculating the RMSD among the crystallized and docked pose. Furthermore, in the caption of docking pictures, authors should reported that the pictures were generated by LigandScout.
-in the discussion, authors claimed that two compounds showed mixed-type inhibition, in the conclusion one compound was claimed to show this behavior. Please clarify.
- Figures in the Supplementary were not cited in the text.
Comments are provided in the major comments section
Reviewer 2 Report
There is an increased need for the development of melanin biosynthesis inhibitors mainly as whitening agents in cosmetics but also for other uses that may include inhibition of browning of vegetables and fruits. The main enzyme that can be targeted to inhibit melanin production is tyrosinase.
Here, the authors build on their own previous data that showed that 2-mercaptobenzo[d]imidazole and 2-mercaptobenzo[d]thiazole derivatives are potent inhibitors of the enzyme tyrosinase, to generate a new class of closely related inhibitors the 2-mercaptomethylimidazole derivatives.
The authors have added various group (electron donative or withdrawing) on the derivatives and determined the IC50 values for tyrosinase inhibition. In total the authors have generated 10 compounds.
The biochemical explanation for tyrosinase inhibition is provided and supported by chelation experiments, enzyme kinetics and in silico (docking) studies. The lead inhibitors exhibit competitive or mixed inhibition.
Further, the compounds did not exhibit toxicity in B16F10 or HaCaT cells, although compound 7 was toxic in zebrafish larvae.
Overall, the study is very interesting and well-designed.
I only have one comment.
Based on their data it appears that the IC50 values does not completely correlate with the melanogenesis in cells or in vivo. Compound 10 has IC50 2x from kojic acid but is better in B16F10 cells and significantly better that kojic acid in vivo in zebrafish model. A discussion on correlation between IC50 and biological effects is missing.
There is an increased need for the development of melanin biosynthesis inhibitors mainly as whitening agents in cosmetics but also for other uses that may include inhibition of browning of vegetables and fruits. The main enzyme that can be targeted to inhibit melanin production is tyrosinase.
Here, the authors build on their own previous data that showed that 2-mercaptobenzo[d]imidazole and 2-mercaptobenzo[d]thiazole derivatives are potent inhibitors of the enzyme tyrosinase, to generate a new class of closely related inhibitors the 2-mercaptomethylimidazole derivatives.
The authors have added various group (electron donative or withdrawing) on the derivatives and determined the IC50 values for tyrosinase inhibition. In total the authors have generated 10 compounds.
The biochemical explanation for tyrosinase inhibition is provided and supported by chelation experiments, enzyme kinetics and in silico (docking) studies. The lead inhibitors exhibit competitive or mixed inhibition.
Further, the compounds did not exhibit toxicity in B16F10 or HaCaT cells, although compound 7 was toxic in zebrafish larvae.
Overall, the study is very interesting and well-designed.
I only have one comment.
Based on their data it appears that the IC50 values does not completely correlate with the melanogenesis in cells or in vivo. Compound 10 has IC50 2x from kojic acid but is better in B16F10 cells and significantly better that kojic acid in vivo in zebrafish model. A discussion on correlation between IC50 and biological effects is missing.
Author Response
We would like to thank the reviewers for their valuable comments and suggestions that further improved the value of our manuscript.
Reviewer 2.
There is an increased need for the development of melanin biosynthesis inhibitors mainly as whitening agents in cosmetics but also for other uses that may include inhibition of browning of vegetables and fruits. The main enzyme that can be targeted to inhibit melanin production is tyrosinase.
Here, the authors build on their own previous data that showed that 2-mercaptobenzo[d]imidazole and 2-mercaptobenzo[d]thiazole derivatives are potent inhibitors of the enzyme tyrosinase, to generate a new class of closely related inhibitors the 2-mercaptomethylimidazole derivatives.
The authors have added various group (electron donative or withdrawing) on the derivatives and determined the IC50 values for tyrosinase inhibition. In total the authors have generated 10 compounds.
The biochemical explanation for tyrosinase inhibition is provided and supported by chelation experiments, enzyme kinetics and in silico (docking) studies. The lead inhibitors exhibit competitive or mixed inhibition.
Further, the compounds did not exhibit toxicity in B16F10 or HaCaT cells, although compound 7 was toxic in zebrafish larvae.
Overall, the study is very interesting and well-designed.
I only have one comment.
Based on their data it appears that the IC50 values does not completely correlate with the melanogenesis in cells or in vivo. Compound 10 has IC50 2x from kojic acid but is better in B16F10 cells and significantly better that kojic acid in vivo in zebrafish model. A discussion on correlation between IC50 and biological effects is missing.
Response:
Thank you for leaving a valuable and detailed review. As per the reviewer's suggestion, in section 3.11, we have added a brief discussion on the correlation between IC50 values for mushroom tyrosinase and biological effects obtained in cells or in vivo.
Round 2
Reviewer 1 Report
authors addressed my concerns
authors addressed my concerns